

# Gene mutation analysis of oral submucous fibrosis cancerization in Hainan Island

Bingxia Li[1,2,*], Xinyu Chen[3,*], Haiyu Xian[1,2], Qitao Wen[1,2] and Tao Wang[1,2]

[1] Department of stomatology, Hainan General Hospital, Haikou, China
[2] The Affiliated Hainan Hospital of Hainan Medical University, Haikou, Hainan, China
[3] College of Stomatology, Hospital of Stomatology, Guangxi Medical University, Nanning, Guangxi, China
* These authors contributed equally to this work.

## ABSTRACT

**Objective:** The sequencing panel composed of 61 target genes was used to explore the related mutation genes of oral squamous cell carcinoma (OSCC) and oral submucous fibrosis (OSF) cancerization, so as to provide a theoretical basis for the early diagnosis of oral submucous fibrosis cancerization, find the most important mutations in OSF cancerization, and more targeted prevention of OSF cancerization.

**Methods:** A total of 74 clinically diagnosed samples were included, including 36 cases of OSCC and 38 cases of OSF cancer patients. DNA was extracted, and targeted gene panel sequencing technology was used to analyze the gene frequency of pathogenic mutation sites in clinical samples.

**Results:** Gene panel sequencing analysis showed that there were 69 mutations in 18 genes in OSCC and OSF cancerous specimens. The results of gene panel sequencing were screened, and 18 mutant genes were finally screened out and their mutation frequencies in the samples were analyzed. According to the frequency of gene mutations from high to low, they were TP53, FLT4, PIK3CA, CDKN2A, FGFR4, HRAS, BRCA1, PTPN11, NF1, KMT2A, RB1, PTEN, MSH2, MLH1, KMT2D, FLCN, BRCA2, APC. The mutation frequency of FLT4 gene was significantly higher than that of OSCC group ($P < 0.05$).

**Conclusion:** FLT4 gene may be related to OSF cancerization and is expected to be an early diagnostic biomarker for OSF cancerization.

## INTRODUCTION

Oral squamous cell carcinoma (OSCC) is the most common malignancy of the head and neck. It is the sixth most common cancer in the world. It seriously affects people's chewing, swallowing and speech functions, it will not only cause physical and psychological damage to patients, but also cause serious personal economic burden (*Chen et al., 2008*; *Epstein et al., 2008*; *Rana et al., 2014*). In our country, there is a lot of cases in Hainan, Hunan, and Taiwan (*Lee et al., 2011*). Residents in these areas have the habit of chewing betel nut, which is the main risk factor related to oral submucous fibrosis (OSF) and its cancerous

Corresponding authors
Xinyu Chen,
chenXinyu814@sr.gxmu.edu.cn
Tao Wang, 18608917377@163.com

transformation into OSCC. The International Agency for Research on Cancer (IARC) has classified areca nut as a Group I carcinogen and has identified it as an independent risk factor for OSCC. Despite the development of comprehensive and multidisciplinary therapies, the prognosis and 5-year survival rate for OSCC remain unsatisfactory, and studies have shown betel quid and areca nut chewing is significantly associated with poor prognosis in patients with OSCC (*Yang et al., 2021*). In the past, some scholars defined OSF cancerization as "oral cancer complicated with or accompanied by OSF", which included cases of leucoplakia cancerization of oral mucosa. After years of research, our team deemed that cases developed from leucoplakia should be excluded and OSF cancerization should be defined as oral squamous cell carcinoma directly transformed by OSF. Because oral cancer is often detected and diagnosed at a later stage, it is usually fatal for those affected, with a death rate of about 80 percent for advanced oral cancer, which can be reduced to about 50 percent with early screening and diagnosis (*Warnakulasuriya, 2009*, *2010*). Therefore, it is essential to identify early diagnostic targets for OSF cancerization.

After decades of development, gene detection technology has formed a complete system, and new technologies are constantly emerging. As a genetic material, DNA is a series of studies on phages from 1930 to 1952, which found the important position of DNA in genetics and confirmed that DNA is the material carrier of genetic information of life. After 30 years of development, the gene sequencing technology platform currently includes first-generation sequencing technology (Sanger sequencing technology), second-generation sequencing technology (high-throughput sequencing technology), and third-generation sequencing technology (single-molecule sequencing technology). Precision medicine has created a whirlwind in the medical community. The prerequisite for precision medicine is accurate diagnosis. It is precisely because of the development of corresponding detection technologies represented by second-generation sequencing technology that the realization of precision medicine has become possible. Gene panel sequencing belongs to the second-generation sequencing technology. Gene panel sequencing can formulate the best treatment plan according to the gene mutation of patients with OSF cancerization and the corresponding clinical conditions, find the potential targeted drugs that patients can benefit to the greatest extent, find the most important mutation in OSF cancerization, and more targeted to prevent the cancerization of OSF. Common types of gene sequencing include whole genome sequencing, whole exon sequencing, transcriptome sequencing, and targeted capture sequencing. Targeted sequencing is a research strategy in which the genome region of interest is enriched by capture kits for sequencing. According to different applications, ultra-high sensitivity and accuracy can be obtained with less data, and rapid screening of variation sites can be realized. Targeted gene panel sequencing is a sequencing strategy and method designed based on whole exome sequencing and whole genome sequencing. Compared with targeted sequencing, targeted sequencing can focus on the region of concern, remove the interference of redundant data, and maximize the use of sequencing reads, which has the advantages of deeper sequencing, lower cost and more sensitivity. It is the preferred method to identify complex disease susceptibility genes at present (*Kamps et al., 2017*).

Single nucleotide variation (SNV) is when one base is replaced by another, also known as single-nucleotide polymorphism (SNP). SNV is generally superior to SNP because only variants found in a single sample are called SNV, whereas SNP is a population concept, and this difference accounts for more than 1% of the population. Therefore, we not only studied SNP, but also studied SNV. No targeted sequencing studies have been found on OSF cancerization. Therefore, here we used a gene sequencing panel composed of oncogenes and tumor suppressor genes to identify the most common and unique mutations in the OSF patients.

## MATERIALS AND METHODS

### Research object and ethical statement

This study included the primary focus tissues of 74 patients with OSCC and OSF cancerization admitted to Hainan General Hospital from June 2020 to September 2021. This study was approved by the Clinical Research Ethics Committee of Hainan General Hospital (Approval No. (2019) 37). All samples used were informed to the patient, and the consent form for sample use was signed before the operation by all patients, of which samples were collected for this research. The research complies with the World Medical Association's Code of Ethics (Helsinki Declaration) for experiments involving human beings. Subjects were recruited according to the defined inclusion and exclusion criteria. For cases: (1) subjects over 18 years old; (2) diagnosed as OSCC or OSF cancerization; (3) all patients with oral cancer were first-time patients without radiotherapy and chemotherapy; (4) patients were residents of the Hainan Province. All tissue samples were stored in the tissue sample storage tube of Kangwei Century within half an hour after the operation, without the influence of formalin fixation or paraffin embedding. The clinical diagnosis of all specimens was confirmed by histopathological examination.

### DNA extraction

Magen tissue DNA extraction kit (D3121; Guangzhou Meiji Biotechnology Co., Ltd., Tokyo, Japan) was used to extract DNA from oral cancer tissue samples, and the steps described in the extraction kit were strictly followed. The DNA concentration was determined using Qubit 3.0 (Thermofly).

### Library construction and gene panel sequencing

Using Hieff NGS OnePot ll DNA Library Prep Kit for MGI (13321ES96; Yeason Biotechnology, Shanghai, China). According to the kit instructions, firstly, 200 ng–500 ng DNA was extracted for enzyme sectioning and terminal repair and dA tail addition at the same time, then the MGI connector was connected, and the connection product was purified by magnetic beads. The purified product was subjected to two rounds of sorting, and the sorted connector connection product was subjected to PCR amplification and enrichment. All operations were performed in strict accordance with the instructions. The construction of the library was completed using Qsep 100 (Guangding Biotechnology, Guangdong, China) for quality inspection. The qualified library was captured by the liquid-phase hybridization capture system of Heyin organism (No. P10006A) and the oral

squamous cell carcinoma panel probe independently designed by Hefei Nowell Gene Technology Service Co., Ltd. Among them, panel probes covered 61 oral squamous cell carcinoma-related genes, including : ABCB1, APC, BRAF, BRCA1, BRCA2, CDA, CDKN2A, CYP19A1, CYP2B6, CYP2C8, CYP2C9, CYP2C19, CYP2D6, CYP3A4, CYP3A5, DHFR, DPYD, EGFR, ERBB2, ERCC1, ERCC2, FGFR4, FLCN, FLT1, FLT4, GSTM1, GSTP1, GSTT1, HRAS, IDH1, IDH2, KDR, KIT, KMT2A, KMT2D, KRAS, MET, MLH1, MSH2, MTHFR, NF1, NQO1 NRAS, NTRK1, NTRK2, NTRK3, PDGFRA, PDGFRB, PIK3CA, PTEN, PTPN11, RB1, RRM1, SLC19A1, SMAD4, SULT1A1, TP53, TPMT, TYMS, UGT1A1, XRCC1. The closed hybridization reaction system was incubated at 65 °C for 4 h on the PCR instrument for hybridization capture, and then the hybridization products were purified by streptomycin affinity magnetic beads, and the unbound DNA was removed by elution to obtain the capture library. Finally, the library was sequenced using BGI's high-throughput sequencing reagent (No. 1000019844) and MGISEQ-200RS high-throughput sequencing platform.

## Experimental grouping and statistical analysis

The subjects were divided into OSCC group and OSF cancerization group for comparison. IBM SPSS Statistics 26.0 software was used for statistical analysis. The measurement data were described by ($\bar{x} \pm s$) and Welch Two Sample t-test was used; The counting data is expressed by passing (%), using X, 2-test; Fisher's exact test is used to detect the association between genes and OSCC; results: the difference was statistically significant with <0.05.

# RESULTS

## Results of tumor specimen collection

All patients with OSCC were selected for surgical resection, and the postoperative pathological diagnosis was OSCC, and the diagnosis was clear. The following are the basic characteristics of 74 cases (Table 1).

## Quality assessment of hybridization capture sequencing data

We performed next-generation sequencing analysis on 36 OSCC samples and 61 genes from 38 OSF cancerization samples classified by histopathology (Table S1). All sample sequencing reads are processed in the following steps: 1. Use trim-galore software to filter low-quality reads and remove adapters to obtain clean date; 2. compare clean date with the reference genome hg19 using the BWA MEN program, and remove PCR repeats during sequencing using samtools and GATK toolkits; 3. analysis of variant sites using the GATK toolkit; 4. annotate mutation sites using VEP, snpEFF, ANNOVAR annotation software, and ClinVar database. All samples have a read length of 100 bp. The summary statistics of the sequencing results show that the total data volume of sample sequencing is 2.97G, the average median target rate is 81.12% (range: 54.52%~98.30%), the average sequencing depth range is 697.41X~9861X, the one-time average target coverage is 98.90% (range: 92.13%~100%), and the average Q30 measurement value is 92.60% (range: 88.37% ~95.73%) (Table S2).

Table 1 Patient characteristics of OSCC (*n* = 74).

| Variable | Figure |
| --- | --- |
| Age (mean) | 55.72 ± 10.728 (34–82) |
| Gender | |
| Male | 52 (70.3%) |
| Female | 22 (29.7%) |
| Primary site | |
| Tongue | 32 |
| Buccal mucosa | 17 |
| Gingiva | 11 |
| Other | 14 |
| TNM stage | |
| I | 7 |
| II | 22 |
| III | 24 |
| IV | 21 |
| Tumor size(cm) | |
| ≤4 (T I/2) | 51 |
| 4 (T 3/4) | 23 |
| Lymph node metastasis | |
| Negative | 40 |
| Positive | 34 |

## Mutation analysis of study subjects

Results by panel sequencing of targeted genes, further bioinformatics analysis revealed 69 mutations (Table S3). Based on the type of mutation, variants are divided into missense mutations (35), nonsense mutations (20), frameshift deletion (13), and frameshift insertion (1) (Fig. 1), The largest number of missense mutations is among them. It can be seen from the gene waterfall plot that FLT4 are all frameshift deletion mutations, and they are all distributed in OSF cancerization samples. Classified by transition/transversion level, a total of 55 mutations were identified, of which 38 were converted and 17 were switched, with a higher number of transitions than reversals among all study subjects, most of which were distributed in mutation rates of 50% or more (Fig. 2). Most mutations were C-to-T transitions (33, including 17 for G>A and 16 for C>T) (Fig 2). *Petljak et al. (2022)* believe that this is due to cytosine mutations produced by overexpression of cytosine deaminase APOBEC3, and the replication of uracil may lead to C to T mutations leading to single-base substitution 2 (SBS2). In addition, the transition of C to T is also associated with breast cancer (*Chou et al., 2017*), lung cancer (*Li et al., 2018*) and other cancers. We also found a large number of C to A transitions, and previous studies have confirmed that it is associated with lung cancer (*Li et al., 2018*).

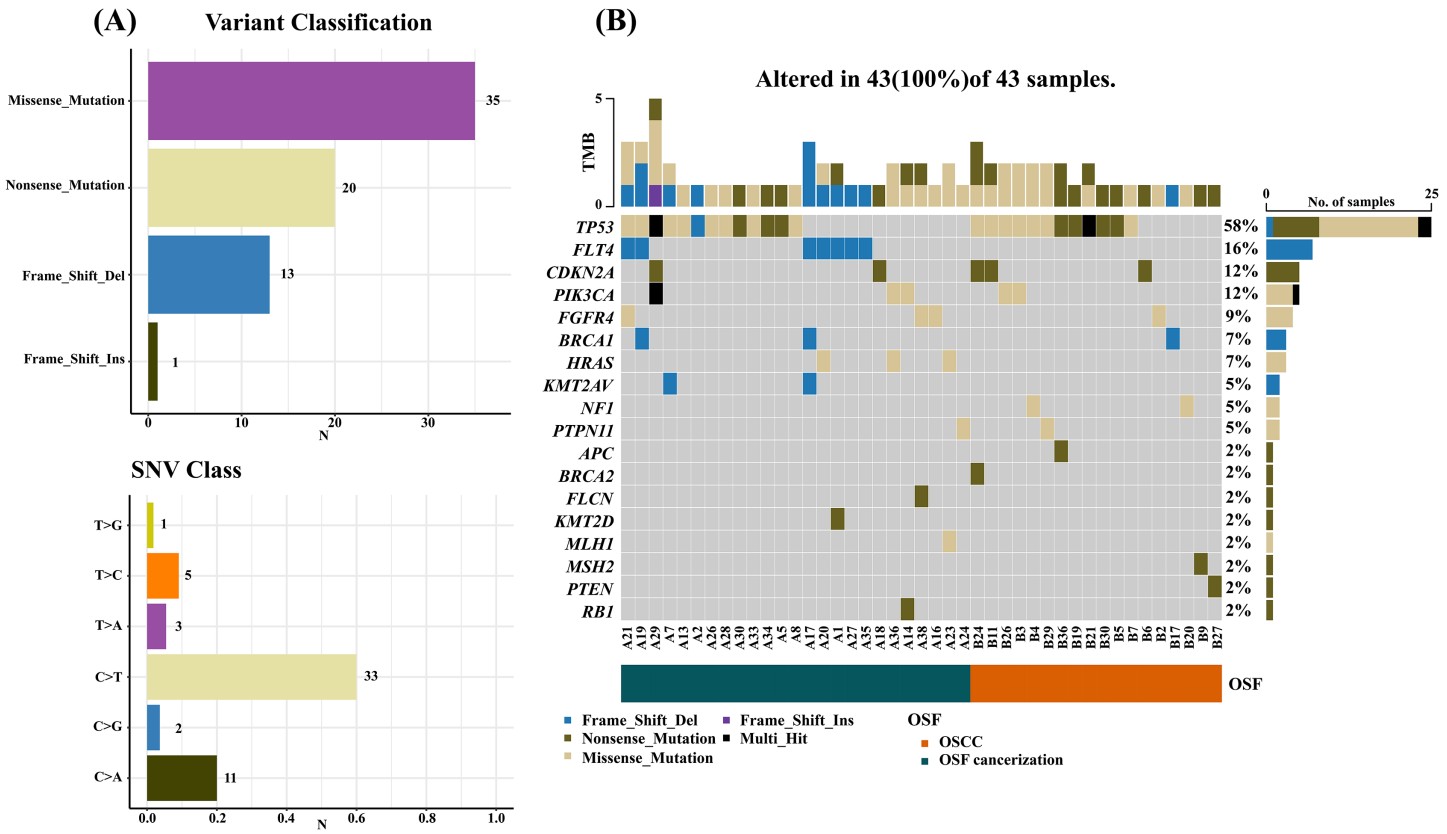

**Figure 1 Mutation summary and gene waterfall plot for SCC-targeted NGS sequencing.** (A) Mutation aggregation. Mutation identification was performed using the "maftools" R program, which summarizes the data generated by NGS and shows the type of mutation and SNV classification, as well as the number of mutations, through a box plot; (B) gene waterfall chart. The figure, generated by the "maftools" R program, represents all types of mutations in highly mutated genes. Comment Multi_Hit: Refers to a gene that has mutated multiple times in the same sample. The figure shows a large number of nonsense and missense mutations in the TP53 gene.                   

## Analysis of sample gene mutation

Gene frequency statistics of pathogenic mutation sites in 74 samples. The mutation screening conditions in each sample were satisfied: Pathogenic on the ClinVar database comment, mutation frequency AF higher than 3%, and GATK software identified the mutation as PASS. Eighteen mutant genes were screened out, and the mutation frequency of these 18 genes in 74 samples was counted, and the mutation frequency of these 18 genes in 74 samples was calculated as TP53, FLT4, PIK3CA, CDKN2A, FGFR4, HRAS, BRCA1, PTPN11, NF1, KMT2A, RB1, PTEN, MSH2, MLH1, KMT2D, FLCN, BRCA2, APC. (Fig. 3). In addition to TP53, the mutant gene of FLT4 gene was the most frequent in 74 oral squamous cell carcinoma samples, The number of mutations was 27 and 7, respectively (Fig. 4).

Seventy-four samples were divided into OSCC group and OSF cancerization group, and the mutation ratio of mutant genes in the two groups was analyzed and compared. Six mutant genes were found to be common to both groups of samples: TP53, PIK3CA, CDKN2A, FGFR4, BRCA1, PTPN11; Five genes were unique to OSCC samples: NF1,

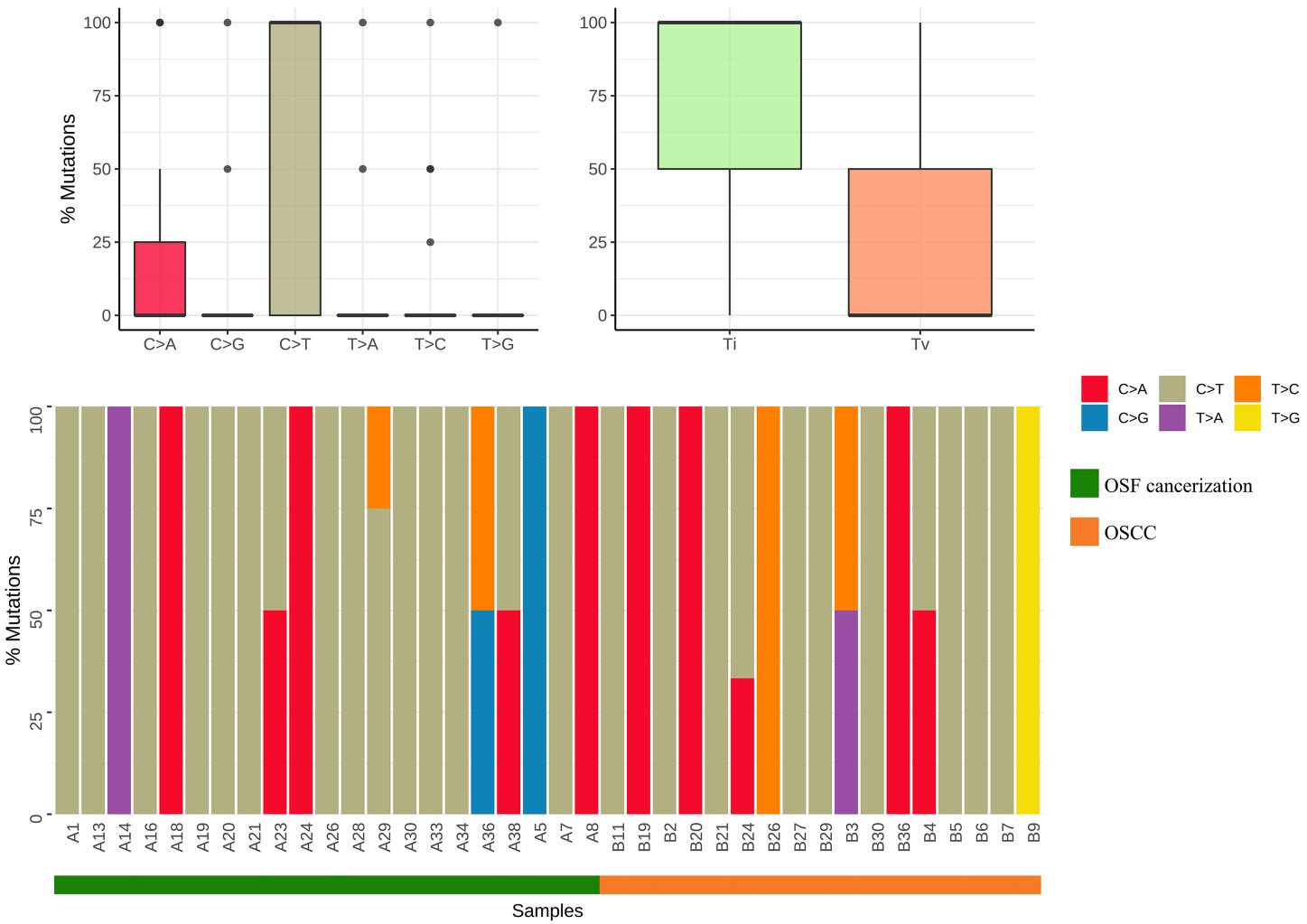

**Figure 2 Mutation classification and proportion of OSCC-targeted NGS sequencing data.** Mutation classification: The "maftools" R procedure is used to identify and classify mutations. The data was visualized to show six different base conversions a boxplot of the population distribution of types, a boxplot of the population distribution of transformations and reversals, and a plot showing the individual base conversion types in each sample stacked column chart of percentages.

PTEN, MSH2, BRCA2, APC; Seven genes were specific to OSF cancerization samples: FLT4, HRAS, KMT2A, RB1, MLH1, KMT2D, FLCN (Table 2).

Mutations were detected in 50% (18/36) of OSCC cases and 65.8% (25/38) of OSF cancerization cases, and the proportion of mutations in the OSF cancerization group was higher than in the OSCC group alone, with 1.444 mutations/tumors and 1.6 mutations/tumors, respectively. The incidence of FLT4 gene mutations in OSF cancerous tissues was 18.4% (7/38), which was significantly higher than that in OSCC tissues alone (0/36) ($P < 0.05$), and the difference was statistically significant (Table 3).

## DISCUSSION

Most patients with oral squamous cell carcinoma are diagnosed in the late stage, resulting in a 5-year survival rate of less than 50% (*Bugshan & Farooq, 2020*). The prognosis is still very poor, so early diagnosis and treatment are still the key to improve the survival rate of

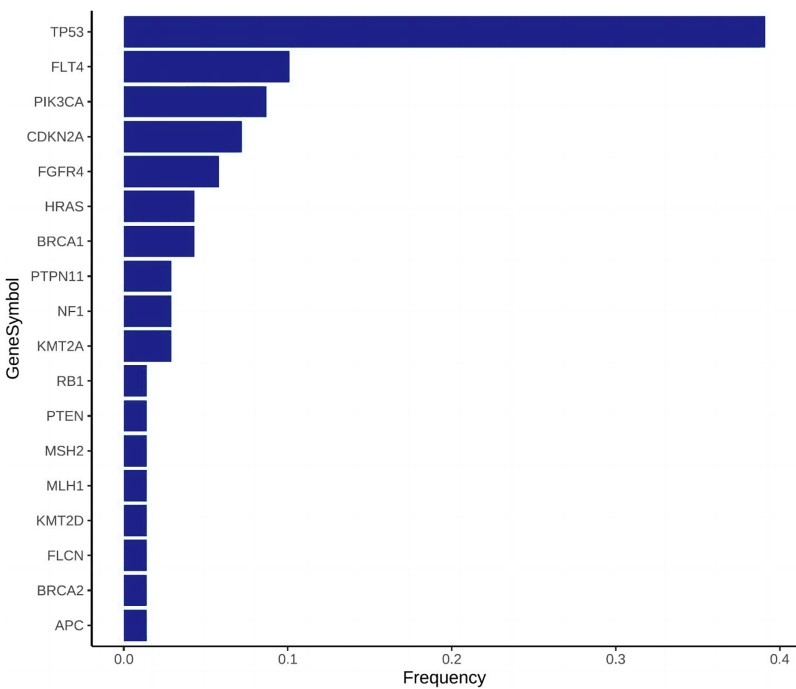

**Figure 3 Mutation frequency bar plot, the abscissa is the mutation frequency in 74 samples, and the ordinate is the gene that has been mutated.**

patients (*Rivera, 2015*). For many years, people have been looking for early tumor biomarkers to find prevention and better treatment options for oral cancer patients. In recent years, various sequencing technologies have found several mutations in oral cancer, including NOTCH1, FAT1, CASP810, *etc*., (*Jayaprakash et al., 2020*), and also identified mutations in the composition of tumor suppressor genes, oncogenes and mitochondrial genes, but no relevant reports have been reported on OSF oncogenic gene mutations. Compared with whole genome and whole exome analysis, there are few articles on gene panel targeted sequencing in oral cancer genome sequencing, which has been used to analyze and detect mutations in breast cancer, ovarian cancer (*Kraus et al., 2017*) and non-small cell lung cancer (*Mosele et al., 2020*). Gene panel targeted sequencing enriches and sequenced the genome region of interest through the capture kit, which enables rapid and cost-effective analysis of mutations in clinical samples and identification at the early stage of cancer occurrence, to achieve "early detection, early diagnosis and early treatment", providing an excellent opportunity for rapid clinical treatment decision-making. Panel targeting sequencing plays a unique role in the new generation of high-throughput sequencing, which has produced many exciting new discoveries and has been applied more and more widely. Therefore, we used Panel targeted sequencing to detect the most common and unique mutations in the OSF cancer group and OSCC group.

In our study, Seven specific gene mutations in the OSF cancer group were identified by gene panel targeted sequencing technology. Seven mutations were identified in FLT4, which was the largest number of mutations, followed by three mutations in HRAS and two mutations in KMT2A. Combining the mutation frequency and distribution of OSCC and

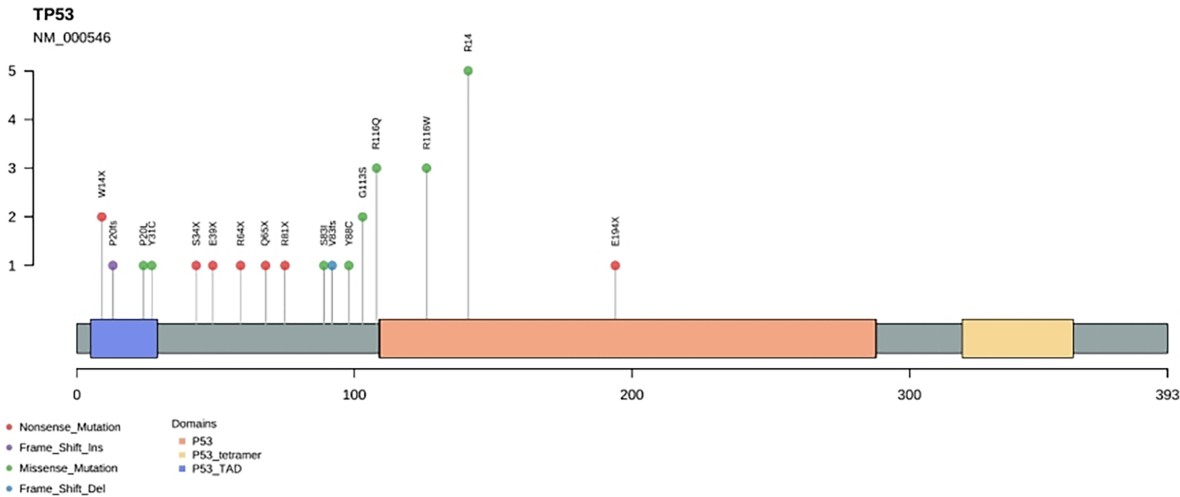

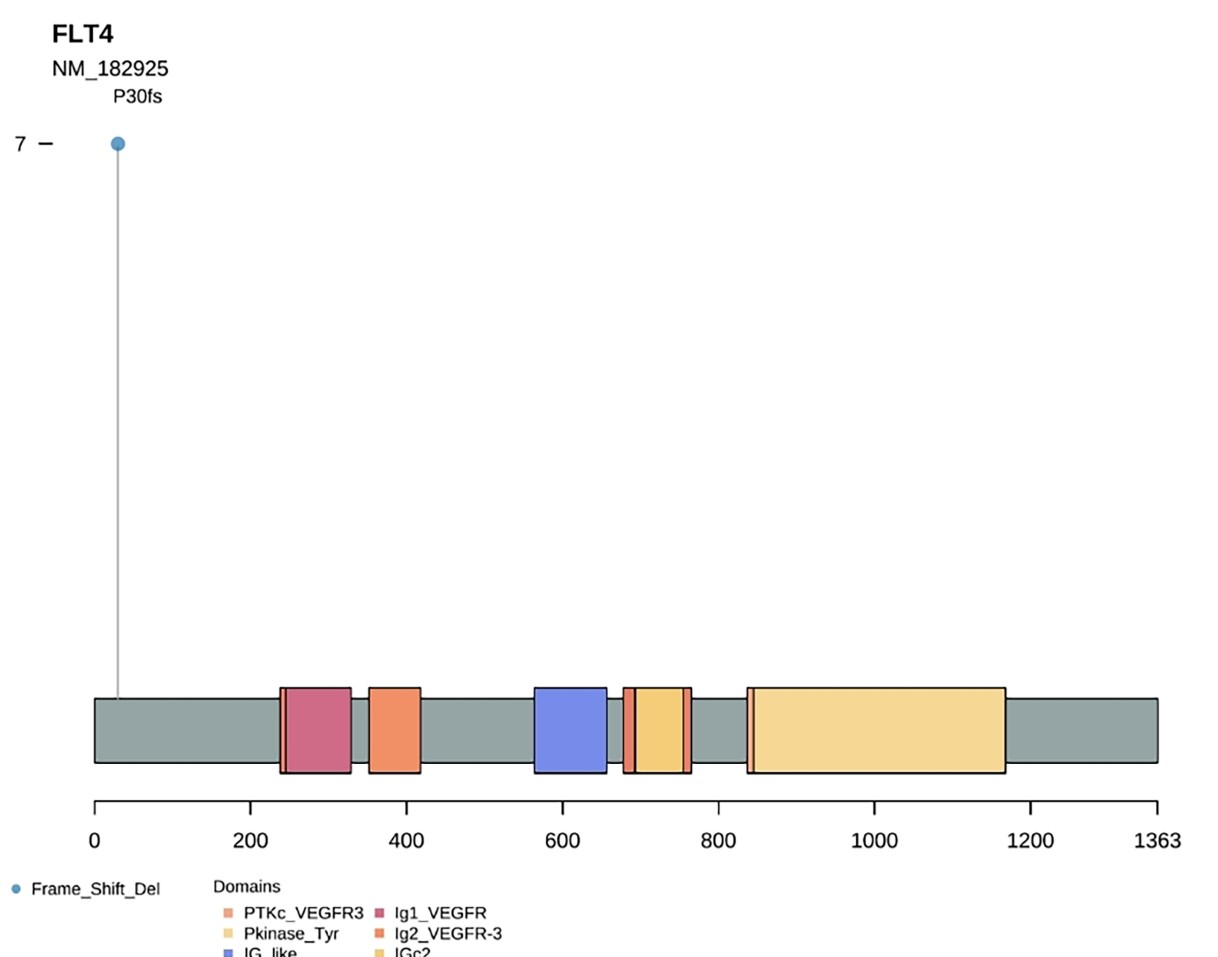

**Figure 4  Needle plot of mutations. Mutations in selected significantly mutated genes across OSCC clinical specimens.**

**Table 2 Distribution of mutated genes in two groups.**

| Gene | OSCC mutated genes | OSF Cancerous mutated genes |
|---|---|---|
| TP53 | √ | √ |
| FLT4 | | √ |
| PIK3CA | √ | √ |
| CDKN2A | √ | √ |
| FGFR4 | √ | √ |
| HRAS | | √ |
| BRCA1 | √ | √ |
| PTPN11 | √ | √ |
| NF1 | √ | |
| KMT2A | | √ |
| RB1 | | √ |
| PTEN | √ | |
| MSH2 | √ | |
| MLH1 | | √ |
| KMT2D | | √ |
| FLCN | | √ |
| BRCA2 | √ | |
| APC | √ | |

**Table 3 Clinicopathological features based on whether OSCC is associated with OSF.**

| | OSCC | OSF canceration | P value |
|---|---|---|---|
| Total | 36(48.6%) | 38(51.4%) | |
| Gender | | | |
| Female | 15 | 7 | 0.05333 |
| Male | 21 | 31 | |
| TNM stage | | | |
| I, II | 13 | 16 | 0.772 |
| III, IV | 23 | 22 | |
| Lymph node metastasis | | | |
| Negative | 21 | 19 | 0.4722 |
| Positive | 15 | 19 | |
| Mutation number | | | |
| 0 | 18 | 13 | 0.2542 |
| ≥1 | 18 | 25 | |
| Mutation number | | | |
| Average | 1.444 | 1.6 | 0.4646 |
| FLT4 mutations | | | |
| Negative | 36 | 31 | 0.02094 |
| Positive | 0 | 7 | |

**Note:**
Number of mutations: the number of mutated genes in each sample, for example: OSCC-mutation number ≥1 is 18: it means that in the group (OSCC), the number of (number of mutated genes per sample ≥1) is 18; except for the t-test used for (mutation average), the rest used the chi-square test; (FLT4 mutation) was accurately tested using Fisher's exact test.

OSF cancer groups, we found that the mutation frequency of FLT4 was significantly higher in OSF cancer group, while the OSCC group had no mutation of FLT4 gene, and the mutation of FLT4 gene was unique to OSF cancer group. The analysis of the relationship between OSCC and FLT4 gene mutation with or without OSF showed that OSCC with or without OSF was correlated with FLT4 gene mutation, and FLT4 gene mutation in OSF cancer group was significantly higher than that in OSCC group alone.

FLT4, also known as vascular endothelial growth factor receptor 3 (VEGFR3), is a highly glycosylated single-chain transmembrane protein. Together with VEGF-C, it is the only group of regulatory factors for lymphangiogenesis in embryonic tissue and the physiological function of lymphangiogenesis in mature individuals (*Kaipainen et al., 1995*). At the same time, it is also associated with cancer and its role in promoting neo angiogenesis (*Gore et al., 2011*). A few studies have reported that the expression level of FLT4 in tumors is significantly positively correlated with the development of cancer cell metastasis and poor prognosis (*Garouniatis et al., 2013*; *Martins et al., 2013*). *Xiao et al. (2015)* found that the increased expression of FLT4 was significantly positively correlated with the invasive tumor phenotype, resulting in poor survival rate, and was associated with lymph node metastasis of colorectal cancer and early and late death of patients.

Panel targeting sequencing found that the mutation frequency of FLT4 in cancer cells of OSCC patients with OSF was extremely high, while that of patients with simple OSCC was almost non-mutated. Chewing areca nuts was the main factor leading to OSF. Therefore, we speculated that, FLT4 gene may be related to OSCC caused by OSF cancerization. FLT4, as a receptor for VEGF-C, passes through lymphatic vessel endothelial cells, promotes lymphatic vessel survival, growth and migration, alters tissue metabolism, attenuates tissue fibrosis and influences collagen metabolism (*Anura et al., 2016*; *Carmeliet, 2000*). We hypothesised that in OSF, mutations in FLT4 could lead to abnormal lymphangiogenesis and collagen accumulation locally in the lesion. With the glass-like degeneration of collagen fibres, the epithelium may atrophy or proliferate or even become cancerous. Meanwhile, due to the obstruction of lymphangiogenesis, the immune cells cannot reach the lesion site fast enough to remove the mutated cells in time, which makes the cancerous lesions easy to arise. At present, there are few studies on the influence and correlation of FLT4 on oral cancer cells. Therefore, in the later stage, we plan to observe whether the physiological function of cancer cells changes and the changes of its related pathways by knocking down the expression of FLT4 in oral cancer cells. In the current study, which used bioinformatics methods to identify candidate biomarkers, we anticipate that the identified FLT4 gene will contribute to our understanding of the molecular mechanisms behind the cancerization of OSF into OSCC and to the identification of new targeted therapies. This experiment also has some limitations with a small sample size, and we will continue to collect cases and expand the sample in subsequent studies with a view to conducting a more comprehensive study.

## CONCLUSIONS

In the present work, we found that the mutation frequency of FLT4 in cancer cells of OSCC patients with OSF was extremely high, while that of patients with OSCC without

OSF was almost non-mutated by panel targeting sequencing. Therefore, we deem that FLT4 gene may be related to OSF cancerization and is expected to be an early diagnostic biomarker for OSF cancerization.

## ACKNOWLEDGEMENTS

The authors gratefully acknowledge all the study participants and study staff for their help and cooperation during this study.

### Funding

This study was supported by the National Natural Science Foundation of China (No. 81960199) and the Hainan Province Science and Technology special Fund (No. ZDYF2021SHFZ114). The funders had no role in study design, data collection and analysis, decision to publish, or preparation of the manuscript.

### Grant Disclosures

The following grant information was disclosed by the authors:
National Natural Science Foundation of China: 81960199.
Hainan Province Science and Technology special Fund: ZDYF2021SHFZ114.

### Competing Interests

The authors declare that they have no competing interests.

### Author Contributions

- Bingxia Li conceived and designed the experiments, analyzed the data, authored or reviewed drafts of the article, and approved the final draft.
- Xinyu Chen conceived and designed the experiments, prepared figures and/or tables, authored or reviewed drafts of the article, and approved the final draft.
- Haiyu Xian performed the experiments, authored or reviewed drafts of the article, and approved the final draft.
- Qitao Wen performed the experiments, authored or reviewed drafts of the article, and approved the final draft.
- Tao Wang conceived and designed the experiments, analyzed the data, authored or reviewed drafts of the article, and approved the final draft.

### Human Ethics

The following information was supplied relating to ethical approvals (*i.e.*, approving body and any reference numbers):

This study was approved by the Clinical Research Ethics Committee of Hainan General Hospital (Approval No. (2019) 37).

## Data Availability

The raw data is available in the Supplemental Files.

## Supplemental Information

Supplemental information for this article can be found online at http://dx.doi.org/10.7717/peerj.16392#supplemental-information.

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
