# Peer review of "Gene mutation analysis of oral submucous fibrosis cancerization in Hainan Island"

_PeerJ, doi:10.7717/peerj.16392_

## Round 0.1 · original submission · Major Revisions

Dear authors,

Please, provide the modifications suggested by the reviewers.

Best regards,
CM

Reviewer 1 ·

Basic reporting

The article in general needs thorough inspection as some sentences need clarity. Please double-check constructions, punctuations, and capitalizations.


Citations are needed in the introduction. For example, "it seriously affects people's chewing, swallowing, and speech functions, and brings huge and personal costs." A citation may be needed, as well as improved sentence construction. "low rate of lymph node metastasis in patients with areca chewing OSCC" also needs a citation.


The objective is quite vague. Expanding more on the roles of the "early targets for OSF canceration" might be helpful to lead the readers to the premise of why it should be discovered.

Please add more emphasis on gene mutations for better understanding.

Experimental design

Any reason why the Magen tissue DNA extraction kit is used? Thank you.

Can this be clarified?-- "It belongs to Hainan province"

What kind of t-test was used? Please clarify.

Please check the errors in the section "Experimental grouping and statistical treatment" -- this heading should also be corrected.

Validity of the findings

How valid is the diagnosis of OSCC? "The diagnosis is clear" is not enough to impose that the specimen is indeed OSCC.

Please expound on the genes that were found to be mutated. If FLT4 is a novel finding, please add more references. As I read the discussion, only 3-4 references were used related to this gene. More citations and interesting points are warranted.

In your conclusion, it was stated that "FLT4 may be related to OSF carcinogenesis and is expected to be an early marker..." If "expected" is used as the term here, please provide more discussion about this.

Additional comments

No additional comments

·

Basic reporting

A little improvement is required in grammar and technical terms
Some more relevant literature should be added.

Experimental design

The research is within the aim and scope of the journal
Research questions are not well defined.
The research gap is addressed.

Validity of the findings

inappropriate and insufficient statistics.
Request to revise the statistical analysis.
Sample size estimation method is not provided.

Conclusions cannot be generalized.
Address the limitations of the study.

·

Basic reporting

Clear and unambiguous, professional English was used throughout.
The article was written in English and had a clear usage of the English, unambiguous, with a technically correct text conforming to professional standards of courtesy and expression. Minor changes were made by this reviewer and are detailed in the Word file of the full paper via marked alterations.
Literature references are scarce and provide insufficient field background/context. More recent literature should be used and up to date content should be informed.
The article included provide no sufficient introduction and background to demonstrate how the work fits into the broader field of knowledge. Relevant and more up to date prior literature should be appropriately referenced.
Structure of the manuscript, figures, tables (in table 3, canceration has to be changed to cancerization, as the present reviewer changed several times in the annotated pdf) seem adequate. Although the authors have attempted to compare their results with the literature in the results section. This should be done only in the discussion. There are mistakes in a similar fashion that are highlighted in the annotated pdf that is uploaded to the authors.
The paper is self-contained with relevant results to hypotheses.

Experimental design

Original primary research within Aims and Scope of the journal was performed.
Research question was well defined, relevant & meaningful.
The submission clearly defined the research question, which was relevant and meaningful. The knowledge gap being investigated was identified, and statements were made as to how the study contributes to filling that gap.
The manuscript methods and results demonstrate that the authors performed rigorous investigation to a high technical level.
Please, confirm that the research has been conducted in conformity with the prevailing ethical standards in the field, attesting that all patients signed the informed consent (as the present reviewer changed at the main manuscript).
Methods most of the times were described with sufficient detail & information to be replicated. Nevertheless, the authors need to clarify what do they mean with “other operations”, for instance, in the library construction and gene panel sequencing section of the material and methods. This would not allow the methods to be reproducible by other investigators. Furthermore, the panel probe of oral squamous cell carcinoma independently designed by Hefei Novel Gene Technology Service Co., Ltd. should also be explained in detail.

Validity of the findings

When the authors demonstrated G to A transition in a large sample size, this should have been explained in detail at the results section, and these results should have been compared with the literature in the discussion section.
The data on which the conclusions are based must also include the limitations of the study, such as the low sample size (usually at the end of the discussion).
The sentences “Single nucleotide variation (SNV) is when one base is replaced by another, also known as single-nucleotide polymorphism (SNP). SNV is generally superior to SNP because only variants found in a single sample are called SNV, whereas SNP is a population concept, and this difference accounts for more than 1% of the population.” should have been included in the introduction section. Also, according to the supplementary table 3, most of the mutations of your study were SNPs. This should also be explained.
Conclusions are well stated, linked to original research question & limited to supporting results.

---

## Round 0.2 · accepted · Accept

After making some improvements, the manuscript has now reached a level of quality that justifies its publication.
Warm regards,
CM

Reviewer 1 ·

Basic reporting

Suggestions were carefully fulfilled. Thank you.

Experimental design

Suggestions were carefully fulfilled. Thank you.

Validity of the findings

Suggestions were carefully fulfilled. Thank you.

Additional comments

None. Thank you.

·

Basic reporting

The authors succeded in their task of major reviewing the original draft.

Experimental design

No comments.

Validity of the findings

The authors succeded in their task of major reviewing the original draft.

Additional comments

The authors succeded in their task of major reviewing the original draft.